# Restricted Generative Projection for One-Class Classification and Anomaly Detection

## Abstract

We present a novel framework for one-class classification and anomaly detection. The core idea is to learn a mapping to transform the unknown distribution of training (normal) data to a known distribution that is supposed to be different from the transformed distribution of unknown abnormal data. Crucially, the target distribution of training data should be sufficiently simple, compact, and informative. The simplicity is to ensure that we can sample from the distribution easily, the compactness is to ensure that the decision boundary between normal data and abnormal data is clear and reliable, and the informativeness is to ensure that the transformed data preserve the important information of the original data. Therefore, we propose to use truncated Gaussian, uniform in hyperball, uniform on hypersphere, or uniform between hyperspheres, as the target distribution. We then minimize the distance between the transformed data distribution and the target distribution while keeping the reconstruction error for the original data small enough. Our model is simple and easy to train especially compared with those based on generative models. Comparative studies on a few benchmark datasets verify the effectiveness of our method in comparison to baselines.

## 1 Introduction

Anomaly detection (AD) aims to distinguish normal data and abnormal data using a model trained on only normal data without using any information of abnormal data (Chandola et al., 2009; Pang et al., 2021; Ruff et al., 2021). AD is useful in numerous real problems such as intrusion detection for video surveillance, fraud detection in finance, and fault detection for sensors. Many AD methods have been proposed in the past decades (Schölkopf et al., 1999; 2001; Tax & Duin, 2004; Liu et al., 2008). For instance, Schölkopf et al. (2001) proposed the one-class support vector machine (OC-SVM) that finds, in a high-dimensional kernel feature space, a hyperplane yielding a large distance between the normal training data and the origin. Tax & Duin (2004) presented the support vector data description (SVDD), which obtains a spherically shaped boundary (with minimum volume) around the normal training data to identify abnormal samples. There are also many deep learning based AD methods (Erfani et al., 2016; Ruff et al., 2018; Golan & El-Yaniv, 2018; Hendrycks et al., 2018; Abati et al., 2019; Pidhorskyi et al., 2018; Zong et al., 2018; Wang et al., 2019; Liznerski et al., 2020; Qiu et al., 2021; Raghuram et al., 2021; Wang et al., 2021).

Deep learning based AD methods may be organized into three categories. The first category is based on compression and reconstruction. These methods usually use autoencoder (Hinton & Salakhutdinov, 2006; Kingma & Welling, 2013) to learn a low-dimensional representation to reconstruct the high-dimensional data (Vincent et al., 2008; Wang et al., 2021). It is expected that the learned autoencoder on the normal training data has a much higher reconstruction error on unknown abnormal data than on normal data. The second category is based on the combination of classical one-class classification (Tax & Duin, 2004; Golan & El-Yaniv, 2018) and deep learning (Ruff et al., 2018; 2019; 2020; Perera & Patel, 2019; Bhattacharya et al., 2021; Shenkar & Wolf, 2022; Chen et al., 2022). For instance, Ruff et al. (2018) proposed a method called deep one-class SVDD. The main idea is to use deep learning to construct a minimum-radius hypersphere to include all the training data, while the unknown abnormal data are expected to fall outside. The last category is based on generative learning or adversarial learning (Malhotra et al., 2016; Deecke et al., 2018; Pidhorskyi et al., 2018; Nguyen et al., 2019; Perera et al., 2019; Goyal et al., 2020; Raghuram et al., 2021; Yan et al., 2021). For example, Perera et al. (2019) proposed to use the generative adversarial net-

work (GAN) (Goodfellow et al., 2014) with constrained latent representation to detect anomalies for image data. Goyal et al. (2020) presented a method called deep robust one-class classification (DROCC). The method aims to find a low-dimensional manifold to accommodate the normal data via an adversarial optimization approach.

Although deep learning AD methods have shown promising performance on various datasets, they still have limitations. For instance, the one-class classification methods such as Deep SVDD (Ruff et al., 2018) only ensure that the normal data could be included by a hypersphere but cannot guarantee that the normal data are distributed evenly in the hypersphere, which may lead to large empty regions in the hypersphere and hence yield incorrect decision boundary. The adversarial learning methods such as (Nguyen et al., 2019; Perera et al., 2019; Goyal et al., 2020) may suffer from high computational cost and instability in optimization.

In this work, we present a restricted generative projection framework for one-class classification and anomaly detection. The model of the framework is efficient to train and able to provide reliable decision boundaries for precise anomaly detection. Our main idea is to train a deep neural network to convert the distribution of normal training data to a target distribution that is simple, compact, and informative, which will provide a reliable decision boundary to identify abnormal data from normal data. There are many choices for the target distribution, such as truncated Gaussian and uniform on hypersphere. Our contributions are three-fold.

- We present a novel framework for one-class classification and anomaly detection. It aims to transform the data distribution to some target distributions that are easy to be violated by unknown abnormal data.

- We present a few simple, compact, and informative target distributions and propose to minimize the distances between the converted data distribution and these target distributions via minimizing the maximum mean discrepancy.

- We conduct extensive experiments to compare the performance of different target distributions and compare our method with state-of-the-art competitors. The numerical results on five benchmark datasets verify the effectiveness of our methods.

## 2 METHODOLOGY

Suppose we have a set of $m$-dimensional training data $\mathbf{X} = \{\mathbf{x}_1, \mathbf{x}_2, \ldots, \mathbf{x}_n\}$ drawn from an unknown bounded distribution $\mathcal{D}_{\mathbf{x}}$ and any samples drawn from $\mathcal{D}_{\mathbf{x}}$ are normal data. We want to train a model $\mathcal{M}$ on $\mathbf{X}$ to determine whether a test data $\mathbf{x}_{\text{new}}$ is drawn from $\mathcal{D}_{\mathbf{x}}$ or not. One may consider to estimate the density function (denoted by $p_{\mathbf{x}}$) of $\mathcal{D}_{\mathbf{x}}$ using some techniques such as kernel density estimation (Rosenblatt, 1956). Suppose the estimation $\hat{p}_{\mathbf{x}}$ is good enough, then one can determine whether $\mathbf{x}_{\text{new}}$ is normal or not according to the value of $\hat{p}_{\mathbf{x}}(\mathbf{x}_{\text{new}})$: if $\hat{p}_{\mathbf{x}}(\mathbf{x}_{\text{new}})$ is zero or close to zero, $\mathbf{x}_{\text{new}}$ is an abnormal data point; otherwise, $\mathbf{x}_{\text{new}}$ is a normal data point [1]. However, the dimensionality of the data is often high and hence it is very difficult to obtain a good estimation $\hat{p}_{\mathbf{x}}$.

We propose to learn a mapping $\mathcal{T} : \mathbb{R}^m \to \mathbb{R}^d$ to transform the unknown bounded distribution $\mathcal{D}_{\mathbf{x}}$ to a known distribution $\mathcal{D}_{\mathbf{z}}$ while there still exists a mapping $\mathcal{T}' : \mathbb{R}^d \to \mathbb{R}^m$ that can recover $\mathcal{D}_{\mathbf{x}}$ from $\mathcal{D}_{\mathbf{z}}$ approximately. Let $p_{\mathbf{z}}$ be the density function of $\mathcal{D}_{\mathbf{z}}$. Then we can determine whether $\mathbf{x}_{\text{new}}$ is normal or not according to the value of $p_{\mathbf{z}}(\mathcal{T}(\mathbf{x}_{\text{new}}))$. To be more precisely, we want to solve the following problem

$$\underset{\mathcal{T},\, \mathcal{T}'}{\text{minimize}}\ \mathcal{M}\left(\mathcal{T}(\mathcal{D}_{\mathbf{x}}), \mathcal{D}_{\mathbf{z}}\right) + \lambda \mathcal{M}\left(\mathcal{T}'(\mathcal{T}(\mathcal{D}_{\mathbf{x}})), \mathcal{D}_{\mathbf{x}}\right), \tag{1}$$

where $\mathcal{M}(\cdot, \cdot)$ denotes some distance metric between two distributions and $\lambda$ is a trade-off parameter for the two terms. Note that if $\lambda = 0$, $\mathcal{T}$ may convert any distribution to $\mathcal{D}_{\mathbf{z}}$ and lose the ability of distinguishing normal data and abnormal data. Based on the universal approximation theorems (Pinkus, 1999; Lu et al., 2017) and substantial success of neural networks, we use deep neural networks (DNN) to model $\mathcal{T}$ and $\mathcal{T}'$ respectively. Let $f_\theta$ and $g_\phi$ be two DNNs with parameters $\theta$ and $\phi$ respectively. We solve the following problem

---

[1]Here we assume that the distributions of normal data and abnormal data do not overlap. Otherwise, it is difficult to determine whether a single point is normal or not.

$$\underset{\theta,\ \phi}{\text{minimize}}\ \mathcal{M}\left(\mathcal{D}_{f_\theta(\mathbf{x})}, \mathcal{D}_\mathbf{z}\right) + \lambda \mathcal{M}\left(\mathcal{D}_{g_\phi(f_\theta(\mathbf{x}))}, \mathcal{D}_\mathbf{x}\right), \tag{2}$$

where $f_\theta$ and $g_\phi$ serve as encoder and decoder respectively. However, problem (2) is intractable because $\mathcal{D}_\mathbf{x}$ is unknown and $\mathcal{D}_{f_\theta(\mathbf{x})}, \mathcal{D}_{g_\phi(f_\theta(\mathbf{x}))}$ cannot be computed analytically. Note that the samples of $\mathcal{D}_\mathbf{x}$ and $\mathcal{D}_{g_\phi(f_\theta(\mathbf{x}))}$ are given and paired. Then the second term in the objective of (2) can be replaced by sample reconstruction error such as $\frac{1}{n}\sum_{i=1}^{n}\|\mathbf{x}_i - g_\phi(f_\theta(\mathbf{x}_i))\|^2$. On the other hand, we can also sample from $\mathcal{D}_{f_\theta(\mathbf{x})}$ and $\mathcal{D}_\mathbf{z}$ easily but their samples are not paired. Hence, the metric $\mathcal{M}$ in the first term of the objective of (2) should be able to measure the distance between two distributions using their finite samples. To this end, we propose to use the kernel maximum mean discrepancy (MMD) (Gretton et al., 2012) to measure the distance between $\mathcal{D}_{f_\theta(\mathbf{x})}$ and $\mathcal{D}_\mathbf{z}$. In statistics, MMD is often used for Two-Sample test and its principle is to find a function that assumes different expectations on two different distributions:

$$\text{MMD}[\mathcal{F}, p, q] = \sup_{\|f\|_\mathcal{H} \leq 1}\left(\mathbb{E}_p[f(\mathbf{x})] - \mathbb{E}_q[f(\mathbf{y})]\right), \tag{3}$$

where $p, q$ are probability distributions, $\mathcal{F}$ is a class of functions $f : \mathbb{X} \to \mathbb{R}$ and $\mathcal{H}$ denotes a reproducing kernel Hilbert space. Its empirical estimate can be given by

$$\begin{aligned}
\text{MMD}^2[\mathcal{F}, X, Y] = &\frac{1}{m(m-1)}\sum_{i=1}^{m}\sum_{j \neq i}^{m}k(\mathbf{x}_i, \mathbf{x}_j) + \frac{1}{n(n-1)}\sum_{i=1}^{n}\sum_{j \neq i}^{n}k(\mathbf{y}_i, \mathbf{y}_j) \\
&- \frac{2}{mn}\sum_{i=1}^{m}\sum_{j=1}^{n}k(\mathbf{x}_i, \mathbf{y}_j),
\end{aligned} \tag{4}$$

where $X = \{\mathbf{x}_1, \ldots, \mathbf{x}_m\}$ and $Y = \{\mathbf{y}_1, \ldots, \mathbf{y}_n\}$ are samples consisting of i.i.d observations drawn from $p$ and $q$, respectively. $k(\cdot, \cdot)$ denotes a kernel function, e.g., $k(\mathbf{x}, \mathbf{y}) = \exp(-\gamma\|\mathbf{x} - \mathbf{y}\|^2)$, a Gaussian kernel.

Based on the above analysis, we reformulate (2) as

$$\underset{\theta,\ \phi}{\text{minimize}}\ \text{MMD}^2(\mathbf{Z}_\theta, \mathbf{Z}_T) + \frac{\lambda}{n}\sum_{i=1}^{n}\|\mathbf{x}_i - g_\phi(f_\theta(\mathbf{x}_i))\|^2, \tag{5}$$

where $\mathbf{Z}_\theta = \{f_\theta(\mathbf{x}_1), f_\theta(\mathbf{x}_2), \ldots, f_\theta(\mathbf{x}_n)\}$ and $\mathbf{Z}_T = \{\mathbf{z}_i : \mathbf{z}_i \sim \mathcal{D}_\mathbf{z},\ i = 1, \ldots, n\}$. The first term of the objective function in (5) makes $f_\theta$ learn the mapping $\mathcal{T}$ from data distribution $\mathcal{D}_\mathbf{x}$ to target distribution $\mathcal{D}_\mathbf{z}$ and the second term ensures that $f_\theta$ can preserve the main information of observations provided that $\lambda$ is sufficiently large. Figure 1 shows the paradigm of our method.

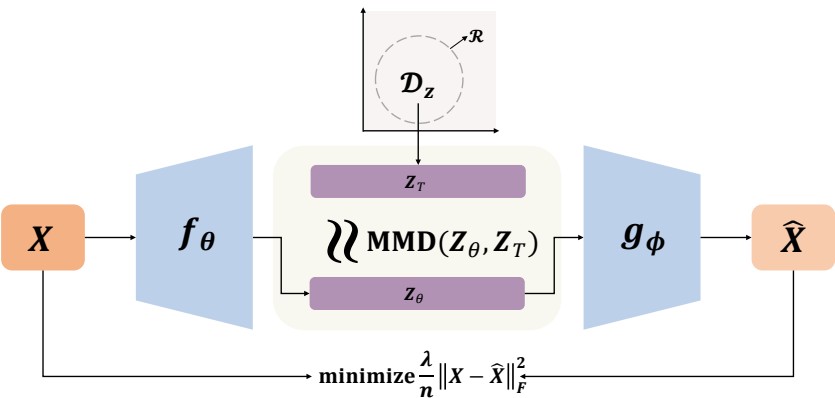

Figure 1: The training stage of Restricted Generative Projection.

Now we introduce four examples of simple and compact $\mathcal{D}_\mathbf{z}$ for (5). The four distributions are Gaussian in Hypersphere (**GiHS**), Uniform in Hypersphere (**UiHS**), Uniform between Hyperspheres (**UbHS**), and Uniform on Hypersphere (**UoHS**). Their 2-dimensional examples are visualized in Figure 2.

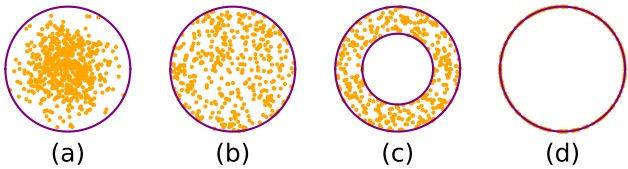

Figure 2: Samples (in orange) from 2-D target (bounded) distributions. Plots (a), (b), (c), (d) denote GiHS, UiHS, UbHS, UoHS respectively.

**GiHS** (Figure 2.a) is actually a truncated Gaussian. Suppose we want to draw $n$ samples from GiHS. A simple approach is drawing $(1+\rho)n$ samples from a standard $d$-dimensional Gaussian and discarding the $\rho n$ samples with larger $\ell_2$ norms. The maximum $\ell_2$ norm of the remaining $n$ points is the radius of the hypersphere. One may also use inverse transform method (Marsaglia, 1963). We have the following proposition (a simple proof can be found in the appendix).

**Proposition 2.1.** *Suppose $\mathbf{z}_1, \mathbf{z}_2, \ldots, \mathbf{z}_n$ are sampled from $\mathcal{N}(\mathbf{0}, \mathbf{I}_d)$ independently. Then for any $r > \sqrt{d/2}$, the following inequality holds*

$$\mathbb{P}\left(\|\mathbf{z}_j\| \geq r\right) \leq \exp\left(-\frac{r^2 + \sqrt{d(2r^2 - d)}}{2}\right), \quad j = 1, \ldots, n. \tag{6}$$

In the proposition, using union bound, we have

$$\mathbb{P}\left(\max_{1 \leq j \leq n} \|\mathbf{z}_j\| \leq r\right) \geq 1 - n\exp\left(-\frac{r^2 + \sqrt{d(2r^2 - d)}}{2}\right). \tag{7}$$

It means a hypersphere of radius $r$ can include all the $n$ samples with probability at least $1 - n\rho$, where $\rho = \exp(-(r^2 + \sqrt{d(2r^2 - d)})/2)$. On the other hand, according to (6), if we sample $n/\rho$ samples from $\mathcal{N}(\mathbf{0}, \mathbf{I}_d)$, the expected number of samples falling into a hypersphere of radius $r$ is at least $n$. Thus, suppose we sample $n'$ samples and remove the $n' - n$ samples with larger $\ell_2$ norms, the expected radius of the smallest hypersphere encasing the remaining $n$ samples is at most $r_0$, where $r_0$ is the solution of the function $\log(n'/n) = r^2 + \sqrt{d(2r^2 - d)})/2$ w.r.t $r$.

**UiHS** (Figure 2.b) is a hyperball in which all the samples are distributed uniformly. To sample from UiHS, we first need to sample from $\mathcal{U}(-1, 1)^d$. Then we discard all the data points outsides the radius-1 hyperball centered at the origin. The following proposition shows some probability result of sampling from a $d$-dimensional uniform distribution.

**Proposition 2.2.** *Suppose $\mathbf{z}_1, \mathbf{z}_2, \ldots, \mathbf{z}_n$ are sampled from $\mathcal{U}(-1, 1)^d$ independently. Then for any $r > 0$, the following inequality holds*

$$\mathbb{P}\left(\|\mathbf{z}_j\| \geq r\right) \leq \frac{4d}{5(3r^2 - d)^2}, \quad j = 1, \ldots, n. \tag{8}$$

Using union bound for (8), we obtain

$$\mathbb{P}\left(\max_{1 \leq j \leq n} \|\mathbf{z}_j\| \leq r\right) \geq 1 - \frac{4dn}{5(3r^2 - d)^2}. \tag{9}$$

It means a hypersphere of radius $r$ can include all the $n$ samples with probability at least $1 - n\rho$, where $\rho = 0.8d^{-1}(3r^2 - d)^2$. On the other hand, (8) indicates that if we draw $n/\rho$ samples from $\mathcal{U}(-1, 1)^d$, the expected number of samples falling into a hypersphere of radius $r$ is at least $n$. Actually, sampling from UiHS is closely related to the Curse of Dimensionality and we need to sample a large number of points from $\mathcal{U}(-1, 1)^d$ if $d$ is large because only a small volume of the hypercube is inside the hyperball. To be more precisely, letting $V_{\text{hypercube}}$ be the volume of a hypercube with length $2r$ and $V_{\text{hyperball}}$ be the volume of a hyperball with radius $r$, we have

$$\frac{V_{\text{hyperball}}}{V_{\text{hypercube}}} = \frac{\pi^{d/2}}{d2^{d-1}\Gamma(d/2)} \triangleq \eta, \tag{10}$$

where $\Gamma$ is the gamma function. Therefore, we need to draw $n/\eta$ samples from $\mathcal{U}(-1, 1)^d$ to ensure that the expected number of samples included in the hyperball is $n$, where $\eta$ is small if $d$ is large.

**UbHS** (Figure 2.c) can be obtained via UiHS. We first sample from UiHS and then remove all samples included by a smaller hypersphere. Since the volume ratio of two hyperballs with radius $r_1$ and $r_2$ is $(\frac{r_1}{r_2})^d$, we need to draw $n/(1 - (r_2/r_1)^d)$ samples from UiHS to ensure that the expected number of samples between the two hyperspheres is $n$.

**UoHS** (Figure 2.d) can be easily obtained via sampling from $\mathcal{N}(\mathbf{0}, \mathbf{I}_d)$. Specifically, for every $\mathbf{z}_i$ drawn from $\mathcal{N}(\mathbf{0}, \mathbf{I}_d)$, we normalize it as $\mathbf{z}_i \leftarrow r\mathbf{z}_i/\|\mathbf{z}_i\|$, where $r$ is the predefined radius of the hypersphere.

In testing stage, we only use the trained $f_\theta^*$ to calculate anomaly scores. For a given test sample $\mathbf{x}_\text{new}$, we define anomaly score $s$ for each target distribution by

$$s(\mathbf{x}_\text{new}) = \begin{cases} \left| \|f_\theta^*(\mathbf{x}_\text{new})\|_2 - r \right|, & \text{for UoHS} \\ \|f_\theta^*(\mathbf{x}_\text{new})\|_2, & \text{for GiHB or UiHS} \\ (\|f_\theta^*(\mathbf{x}_\text{new})\|_2 - r_1) \cdot (\|f_\theta^*(\mathbf{x}_\text{new})\|_2 - r_2), & \text{for UbHS} \end{cases} \quad (11)$$

We call our method Restricted Generative Projection (RGP), which has four variants, denoted by **RGP-GiHS**, **RGP-UiHS**, **RGP-UbHS**, and **RGP-UoHS** respectively, though any target bounded distribution applies.

## 3    CONNECTION WITH PREVIOUS WORK

Our method has a connection with the variational autoencoder (VAE) (Kingma & Welling, 2013). Both methods are autoencoders. The latent distribution in VAE is often Gaussian and not bounded while the latent distribution in our method is more general and bounded. The optimizations of VAE and our method are also different: VAE involves KL-divergence while our method involves MMD.

Our method is closely related to Deep SVDD (Ruff et al., 2018). Both our method and Deep SVDD aim to project the normal training data into some space such that a decision boundary between normal data and unknown abnormal data can be found easily. Deep SVDD minimizes the sum of the squared distances between the projected data and a predefined center to find a hypersphere to include the normal training data, which cannot ensure that the data in the hypersphere are evenly distributed or close to Gaussian and hence may lead to an ineffective decision boundary.

Compared with the autoencoder based anomaly detection method NAE (Yoon et al., 2021) that uses reconstruction error to normalize autoencoder, our method pays more attention to learning a mapping that can transform the unknown data distribution into a simple and compact target distribution.

Similar to our method, Perera et al. (2019) also considered bounded latent distribution in autoencoder for anomaly detection. They proposed to train a denoising autoencoder with a hyper-cube (multi-dimensional uniform) supported latent space, via adversarial training. Obviously, the latent distribution and optimization of our method are different from theirs. In addition, the latent distributions of our method are more compact than the multi-dimensional uniform latent distribution of their method.

## 4    EXPERIMENTS

### 4.1    DATASETS AND BASELINES

In this section, we compare the proposed method with several state-of-the-art anomaly detection methods on three tabular datasets and two widely used image datasets for one-class classification. The datasets are detailed as follows.

- **Abalone** (Dua, 2017) is a dataset of physical measurements of abalone to predict the age. It contains 1,920 instances with 8 attributes.
- **Arrhythmia** (Rayana, 2016) is an ECG dataset. It was used to identify arrhythmic samples in five classes and contains 452 instances with 279 attributes.

- **Thyroid** (Rayana, 2016) is a hypothyroid disease dataset that contains 3,772 instances with 6 attributes.

- **Fashion-MNIST** (Xiao et al., 2017) contains 70,000 grey-scale images with 10 classes.

- **CIFAR-10** (Krizhevsky et al., 2009) is a widely-used benchmark for image anomaly detection. It contains 60,000 color images with 10 classes.

We compare our method with three classic shallow models, four deep autoencoder based methods, three deep generative model based methods, and some latest anomaly detection methods.

- **Classic shallow models**: local outlier factor(LOF) (Breunig et al., 2000), one-class support vector machine(OC-SVM) (Schölkopf et al., 2001), isolation forest (IF) (Liu et al., 2008).

- **Deep autoencoder based methods**: denoising auto-encoder (DAE) (Vincent et al., 2008), DCAE (Seeböck et al., 2016), E2E-AE and DAGMM (Zong et al., 2018), DCN (Caron et al., 2018).

- **Deep generative model based methods**: AnoGAN(Schlegl et al., 2017), ADGAN(Deecke et al., 2018), OCGAN (Perera et al., 2019),

- **Some latest anomaly detection methods**: DeepSVDD(Ruff et al., 2018), GOAD (Bergman & Hoshen, 2020), DROCC (Goyal et al., 2020), HRN (Hu et al., 2020), SCADN (Yan et al., 2021), NeuTraL AD (Qiu et al., 2021), GOCC (Shenkar & Wolf, 2022).

### 4.2 IMPLEMENTATION DETAILS AND EVALUATION METRICS

In this section, we introduce the implementation details of the proposed method RGP and describe experimental settings for image and tabular datasets. Note that all the compared methods do not utilize any pre-trained feature extractors.

For the three tabular datasets (Abalone, Arrhythmia, and Thyroid), in our method, $f_\theta, g_\phi$ are both MLPs. We follow the dataset preparation of (Zong et al., 2018) to preprocess the three tabular datasets for one-class classification task. The hyper-parameter $\lambda$ is set to 1.0 for the three datasets. For the two image datasets (Fashion-MNIST and CIFAR-10), in our method, $f_\theta, g_\phi$ are both CNNs. Since both image datasets contain 10 different classes, we conduct 10 independent one-class classification tasks on both datasets. In each task on CIFAR-10, there are 5,000 training samples and 10,000 testing samples. In each task on Fashion-MNIST, there are 6,000 training samples and 10,000 testing samples. The hyper-parameter $\lambda$ is chosen from $\{0.05, 0.2, 0.1, 1.0\}$ and varies for different classes.

In our method, regarding the restricted target distributions GiHS, we first generate a large number (denoted by $N$) of samples from Gaussian or uniform, sort the samples according to their $\ell_2$ norms, and set $r$ to be the $pN$-th smallest $\ell_2$ norm, where $p = 0.9$. In each iteration (mini-batch) of the optimization, we resample $\mathbf{Z}_T$ according to $r$. The sampling for UiHS and UbHS are similar to GiHS and hence omitted here for simplicity. For UoHS, we draw samples from Gaussian and normalize them to have unit $\ell_2$ norm, then they lie on a unit hypersphere uniformly. The procedure is repeated in each iteration (mini-batch) of the optimization. We use Adam (Kingma & Ba, 2014) as the optimizer in our method. For Fashion-MNIST, CIFAR-10, and Arrhythmia, the learning rate is set to 0.0001. For Abalone and Thyroid, the learning rate is set to 0.001. The details of our network settings are provided in the supplementary material. All experiments were run on AMD EPYC CPU with 64 cores and with NVIDIA Tesla A100 GPU, CUDA 11.6.

To evaluate the performance of all methods, we follow the previous works such as (Ruff et al., 2018) and (Zong et al., 2018) to use AUC (Area Under the ROC curve) for image datasets and F1-score for tabular datasets.

### 4.3 RESULTS ON IMAGE DATASETS

Table 1 and Table 2 compare RGP under four different settings with classic shallow methods, deep autoencoder based methods, generative model based methods, and some latest state-of-the-art methods. We have the following observations.

- Firstly, in contrast to classic shallow methods such as OC-SVM(Schölkopf et al., 2001) and IF (Liu et al., 2008), our RGP has significantly higher AUC scores on all classes of CIFAR-10 and most classes of Fashion-MNIST. An interesting phenomenon is that most deep learning based methods including ours have inferior performance compared to IF (Liu et al., 2008) on class 'Sandal' of Fashion-MNIST.

- Our methods outperformed the deep autoencoder based methods and generative model based methods in most cases and have competitive performance compared to the state-of-the-art in all cases.

- RGP has superior performance on most classes of Fashion-MNIST and CIFAR-10 datasets under the setting of UoHS(uniform distribution on hypersphere). Furthermore, our four settings have relatively close performance on Fashion-MNIST. On CIFAR-10, UoHS outperformed other settings consistently.

Table 1: Average AUC(%) of one-class anomaly detection on Fashion-MNIST. For the competitive methods we only report their mean performance due to the space limit, while we further report the standard deviation for the proposed methods. * denotes we run the official released code to obtain the results, and the best two results are marked in **bold**.

| Normal Class | T-shirt | Trouser | Pullover | Dress | Coat | Sandal | Shirt | Sneaker | Bag | Ankle-boot |
|---|---|---|---|---|---|---|---|---|---|---|
| OC-SVM (Schölkopf et al., 2001) | 86.10 | 93.90 | 85.60 | 85.90 | 84.60 | 81.30 | 78.60 | 97.60 | 79.50 | 97.80 |
| IF (Liu et al., 2008) | 91.00 | 97.80 | 87.20 | **93.20** | 90.50 | **93.00** | **80.20** | 98.20 | 88.70 | 95.40 |
| DAE (Vincent et al., 2008) | 86.70 | 97.80 | 80.80 | 91.40 | 86.50 | **92.10** | 73.80 | 97.70 | 78.20 | 96.30 |
| DAGMM (Zong et al., 2018) | 42.10 | 55.10 | 50.40 | 57.00 | 26.90 | 70.50 | 48.30 | 83.50 | 49.90 | 34.00 |
| ADGAN (Deecke et al., 2018) | 89.90 | 81.90 | 87.60 | 91.20 | 86.50 | 89.60 | 74.30 | 97.20 | 89.00 | 97.10 |
| OCGAN (Perera et al., 2019) | 85.50 | 93.40 | 85.00 | 88.10 | 85.80 | 88.50 | 77.50 | 93.90 | 82.70 | 97.80 |
| DeepSVDD (Ruff et al., 2018) | 79.10 | 94.00 | 83.00 | 82.90 | 87.00 | 80.30 | 74.90 | 94.20 | 79.10 | 93.20 |
| DROCC* (*Goyal et al.*, 2020) | 88.32 | **97.94** | 87.31 | 87.89 | 86.53 | 91.80 | 77.64 | 95.37 | 81.35 | 94.75 |
| HRN (Hu et al., 2020) | 92.70 | **98.50** | 88.50 | 93.10 | **92.10** | 91.30 | 79.80 | **99.00** | **94.60** | **98.80** |
| RGP-GiHS (Ours) | **93.37** (0.70) | 97.37 (0.20) | 89.03 (0.02) | 92.63 (0.31) | **92.41** (0.11) | 87.37 (2.38) | 79.42 (0.86) | 98.09 (0.54) | 91.36 (0.16) | 96.86 (0.03) |
| RGP-UiHS (Ours) | 90.49 (0.09) | 97.19 (0.29) | **90.88** (0.18) | 91.26 (0.72) | 91.26 (0.10) | 89.37 (0.12) | 79.97 (0.15) | 97.57 (0.06) | **92.02** (1.11) | 97.82 (0.36) |
| RGP-UbHS (Ours) | 91.10 (0.28) | 97.20 (0.15) | 89.90 (0.21) | 92.29 (0.15) | 91.35 (0.31) | 82.33 (5.27) | 78.59 (1.53) | 96.98 (0.62) | 91.23 (1.02) | 96.55 (1.12) |
| RGP-UoHS (Ours) | **92.84** (0.41) | 97.23 (0.15) | **89.99** (0.32) | **93.25** (0.35) | 90.73 (0.40) | 85.09 (1.15) | **80.02** (1.05) | **98.54** (0.18) | 91.14 (0.26) | **97.99** (0.29) |

Table 2: Average AUC(%) of one-class anomaly detection on CIFAR-10. For the competitive methods we only report their mean performance due to the space limit, while we further report the standard deviation for the proposed method. * denotes we run the official released code to obtain the results, and the best two results are marked in **bold**.

| Normal Class | Airplane | Auto-mobile | Bird | Cat | Deer | Dog | Frog | Horse | Ship | Trunk |
|---|---|---|---|---|---|---|---|---|---|---|
| OC-SVM (Schölkopf et al., 2001) | 61.10 | 63.80 | 50.00 | 55.90 | 66.00 | 62.40 | 74.70 | 62.60 | 74.90 | 75.90 |
| IF (Liu et al., 2008) | 66.10 | 43.70 | **64.30** | 50.50 | 74.30 | 52.30 | 70.70 | 53.00 | 69.10 | 53.20 |
| DCAE (Seebōck et al., 2016) | 59.10 | 57.40 | 48.90 | 58.40 | 54.00 | 62.20 | 51.20 | 58.60 | 76.80 | 67.30 |
| DAE (Vincent et al., 2008) | 41.10 | 47.80 | 61.60 | 56.20 | **72.80** | 51.30 | 68.80 | 49.70 | 48.70 | 37.80 |
| DAGMM (Zong et al., 2018) | 41.40 | 57.10 | 53.80 | 51.20 | 52.20 | 49.30 | 64.90 | 55.30 | 51.90 | 54.20 |
| AnoGAN-(Schlegl et al., 2017) | 67.10 | 54.70 | 52.90 | 54.50 | 65.10 | 60.30 | 58.50 | 62.50 | 75.80 | 66.50 |
| ADGAN (Deecke et al., 2018) | 63.20 | 52.90 | 58.00 | 60.60 | 60.70 | 65.90 | 61.10 | 63.00 | 74.40 | 64.20 |
| OCGAN (Perera et al., 2019) | 75.70 | 53.10 | 64.00 | 62.00 | 72.30 | 62.00 | 72.30 | 57.50 | 82.00 | 55.40 |
| DeepSVDD (Ruff et al., 2018) | 61.70 | 65.90 | 50.80 | 59.10 | 60.90 | 65.70 | 67.70 | **67.30** | 75.90 | 73.10 |
| DROCC* (*Goyal et al.*, 2020) | **80.10** | **73.41** | **68.78** | 63.36 | 70.81 | 65.01 | 68.83 | **71.13** | 63.81 | 75.49 |
| HRN (Hu et al., 2020) | 77.30 | **69.90** | 60.60 | **64.40** | 71.50 | 67.40 | **77.40** | 64.90 | **82.50** | **77.30** |
| RGP-GiHS (Ours) | 71.88 (0.24) | 63.69 (0.65) | 57.06 (0.36) | 59.12 (1.86) | 67.73 (1.69) | 61.29 (1.19) | 74.52 (0.60) | 63.61 (0.82) | 76.40 (0.41) | 73.36 (0.58) |
| RGP-UiHS (Ours) | 69.83 (0.47) | 64.17 (1.17) | 63.69 (2.03) | 59.55 (1.49) | 67.41 (0.90) | **67.90** (1.06) | 77.34 (0.98) | 64.01 (1.13) | 78.73 (0.11) | 75.08 (1.36) |
| RGP-UbHS (Ours) | 66.96 (4.14) | 66.27 (0.75) | 61.00 (0.09) | 57.34 (2.25) | 68.47 (1.23) | 67.29 (0.06) | 76.33 (0.74) | 63.87 (0.02) | 77.13 (0.29) | 72.63 (0.12) |
| RGP-UoHS (Ours) | **79.30** (0.50) | 68.05 (0.02) | 62.93 (0.96) | **64.64** (0.05) | **73.85** (0.09) | 68.32 (0.20) | **80.08** (0.32) | 66.29 (0.19) | **82.23** (0.21) | **77.56** (0.74) |

## 4.4 Results on Tabular Datasets

In Table 3, we report the F1-scores of our methods in comparison to ten baselines on Arrhythmia, Thyroid, and Abalone. Our four methods significantly outperform all baseline methods in all cases.

Particularly, RGP-UoHS has $18.73\%$, $8.70\%$, and $16.96\%$ improvements on the three datasets in terms of F1-score compared to the runner-up, respectively. Compared with the performance improvement of RGPs on Fashion-MNIST and CIFAR-10, RGPs on the three tabular datasets are more significant.

In addition to the quantitative results, we choose Thyroid (with 6 attributes) as an example and transform the data distribution to 2-dimensional target distributions, which are visualized in Figure 3. We see that RGPs are effective to transform the data distribution to the restricted target distributions. Therefore, RGPs can learn clear boundaries for normal data and hence have satisfactory performance in anomaly detection.

Table 3: Average F1-Scores(%) with standard deviation on three tabular datasets.* denotes we run the official released code of NeuTral AD to obtain the result of Abalone, and the results of Arrhythmia and Thyroid are from the original paper (Qiu et al., 2021). The best two results are marked in **bold**.

| Methods | Abalone | Arrhythmia | Thyroid |
|---|---|---|---|
| OC-SVM (Schölkopf et al., 2001) | $48.00 \pm 0.00$ | $46.00 \pm 0.00$ | $39.00 \pm 1.00$ |
| LOF (Breunig et al., 2000) | $33.00 \pm 1.00$ | $51.00 \pm 1.00$ | $54.00 \pm 1.00$ |
| DCN (Caron et al., 2018) | $40.00 \pm 1.00$ | $38.00 \pm 3.00$ | $33.00 \pm 3.00$ |
| E2E-AE (Zong et al., 2018) | $33.00 \pm 3.00$ | $45.00 \pm 3.00$ | $13.00 \pm 4.00$ |
| DAGMM (Zong et al., 2018) | $20.00 \pm 3.00$ | $49.00 \pm 3.00$ | $49.00 \pm 4.00$ |
| DeepSVDD (Ruff et al., 2018) | $62.00 \pm 1.00$ | $54.00 \pm 1.00$ | $73.00 \pm 0.00$ |
| GoAD (Bergman & Hoshen, 2020) | $61.00 \pm 2.00$ | $51.00 \pm 2.00$ | $72.00 \pm 1.00$ |
| DROCC (Goyal et al., 2020) | $68.00 \pm 2.00$ | $69.00 \pm 2.00$ | $78.00 \pm 3.00$ |
| NeuTral AD* ($Qiu\ et\ al.$, 2021) | $62.07 \pm 2.81$ | $60.30 \pm 1.10$ | $76.80 \pm 1.90$ |
| GOCC (Shenkar & Wolf, 2022) | - | $61.80 \pm 1.80$ | $76.80 \pm 1.20$ |
| RGP-GiHS (Ours) | $84.79 \pm 0.36$ | $\mathbf{78.63 \pm 0.55}$ | $\mathbf{92.60 \pm 0.37}$ |
| RGP-UiHS (Ours) | $\mathbf{84.79 \pm 0.65}$ | $76.98 \pm 0.72$ | $91.83 \pm 0.39$ |
| RGP-UbHS (Ours) | $83.79 \pm 0.81$ | $74.22 \pm 0.78$ | $89.53 \pm 0.00$ |
| RGP-UoHS (Ours) | $\mathbf{86.73 \pm 1.10}$ | $\mathbf{77.70 \pm 0.40}$ | $\mathbf{94.96 \pm 0.28}$ |

Figure 3: Visualization of mapping data distribution to 2-dimensional target distribution on Thyroid datasets. Plots (a), (b), (c), (d) refer to **GiHS**, **UiHS**, **UbHS**, **UoHS**, respectively. The blue points, orange points, green points, red points denote samples from target distribution, samples from training data, normal samples from testing set and abnormal samples from testing set, respectively.

## 4.5 ABLATION STUDY

There is one hyperparameter in our method, namely $\lambda$ in problem (5). Now we show the influence of $\lambda$ on the performance of our method. Figure 4 shows F1-scores of our methods with $\lambda$ varying

from 0 to 100, on the three tabular datasets. Too small or too large $\lambda$ can lower the performance of RGP. When $\lambda$ is very tiny, the reconstruction term of (5) makes less impact on the training target and $f_\theta$ can easily transform the training data to the target distribution but ignores the importance of original data distribution (see Figure 5). On the other hand, when $\lambda$ is very large, the MMD term of optimization objective becomes trivial for the whole training target and $f_\theta$ under the constraint of reconstruction term more concentrates on the original data distribution yet can not learn a good mapping from data distribution to target distribution. Figure 5 illustrates the influence of hyperparameter $\lambda$ on training set of Thyroid dataset. It can be clearly observed that $f_\theta$ transform training data to target distribution better with the decrease of the $\lambda$.

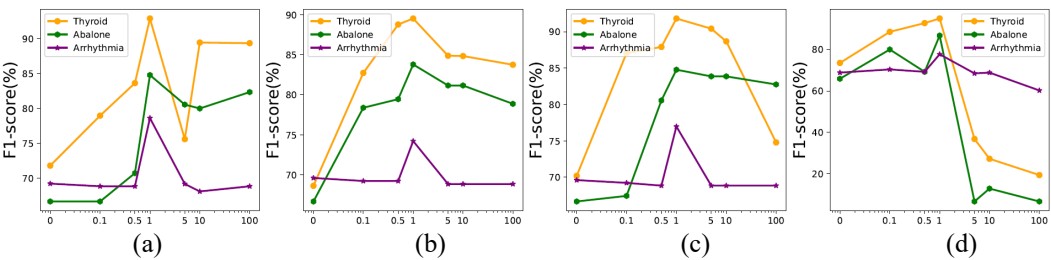

Figure 4: The ablation study of hyper-parameter $\lambda$ on testing set of three tabular datasets under four different restrictions. Plots (a), (b), (c), (d) correspond to **GiHS**, **UiHS**, **UbHS**, **UoHB**, respectively. $\lambda$ is chosen from $\{0, 0.1, 0.5, 1, 5, 10, 100\}$

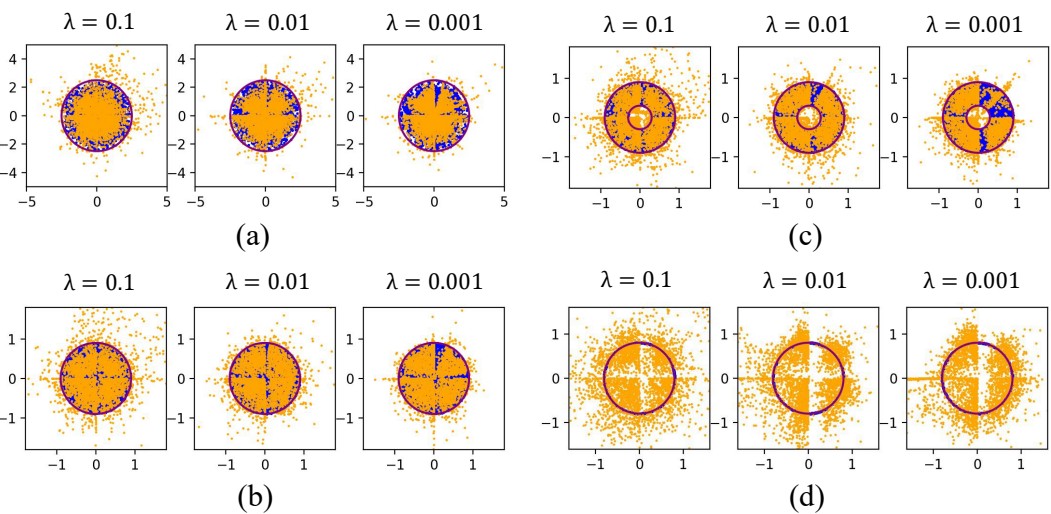

Figure 5: The ablation study of hyper-parameter $\lambda$ on training set of Thyroid dataset under four different restrictions. Plots (a), (b), (c), (d) correspond to **GiHS**, **UiHS**, **UbHS**, **UoHS**, respectively.

## 5 CONCLUSION

We have presented a novel and simple framework for one-class classification and anomaly detection. Our method RGP aims to convert the data distribution to a simple, compact and informative target distribution that can be easily violated by abnormal data. We presented four target distributions and the numerical results showed that uniform on hypersphere is more effective than other distributions in anomaly detection. The reason is that the uniform on hypersphere is much more compact than other distributions and hence provides smaller room for abnormal data points to fall into.

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

## A   MORE NUMERICAL RESULTS

In addition to the results of Tables, we here report the average AUC of all classes in Figure 6. We see that on Fashion-MNIST, our methods have competitive performance as IF and HRN. On CIFAR-10, our RGP-UoHS outperformed all other methods.

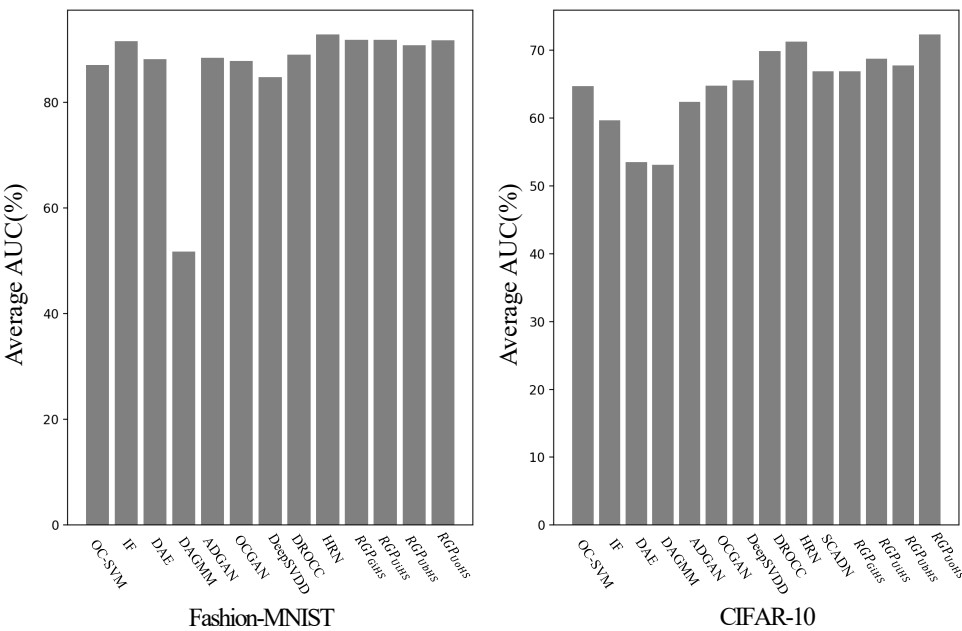

Figure 6: The Average AUC(%) score of Fashion-MNIST and CIFAR-10 on all classes.

We also conduct one-class anomaly detection on MNIST, and report experimental results in Table 4.

Table 4: Average AUC(%) of one-class anomaly detection on MNIST.

| Normal Class | 0 | 1 | 2 | 3 | 4 | 5 | 6 | 7 | 8 | 9 | Avg |
|---|---|---|---|---|---|---|---|---|---|---|---|
| OC-SVM (Schölkopf et al., 2001) | 98.60 | 99.50 | 82.50 | 88.10 | 94.90 | 77.10 | 96.50 | 93.70 | 88.90 | 93.10 | 91.28 |
| IF (Liu et al., 2008) | 98.00 | 97.30 | 88.60 | 89.90 | 92.70 | 85.50 | 95.60 | 92.00 | 89.90 | 93.50 | 92.29 |
| DCAE (Seeböck et al., 2016) | 97.60 | 98.30 | 85.40 | 86.70 | 86.50 | 78.20 | 94.60 | 92.30 | 86.50 | 90.40 | 89.64 |
| ANOGAN (Deecke et al., 2018) | 96.60 | 99.20 | 85.00 | 88.70 | 89.40 | 88.30 | 94.70 | 93.50 | 84.90 | 92.40 | 91.27 |
| Deep SVDD (Ruff et al., 2018) | 98.00 | **99.70** | 91.70 | 91.90 | 94.90 | 88.50 | **98.30** | 94.60 | **93.90** | 96.50 | 94.79 |
| RGP-UoHS (Ours) | **99.50** (0.07) | 98.90 (0.27) | **94.81** (0.59) | **94.89** (0.28) | **95.55** (0.32) | **92.05** (1.75) | 97.41 (0.14) | **96.39** (0.21) | 90.95 (0.35) | 96.25 (0.24) | **95.66** |

In addition to comparison of evaluation metrics, we also measure the time cost of different types of anomaly detection methods on training phase with GUP acceleration or not. We run the experiments of one-class classification on MNIST, Fashion-MNIST, CIFAT10 and report the time cost in Table 5, 6, 7. Table 5 compares training time of Deep SVDD (Ruff et al., 2018), DROCC (Goyal et al., 2020), OCGAN (Perera et al., 2019), RGP on MNIST with the same batch size and training epochs, and without GPU acceleration. Table 6 and Table 7 show the training time of Deep SVDD (Ruff et al., 2018), DROCC (Goyal et al., 2020), RGP on Fashion-MNIST, CIFAR10 with accelerating of NVIDIA RTX2080 GPU(1x). Table 6 and Table 7 indicate that the three methods have the same magnitude on time cost with GPU acceleration and our method has a few advantage on both Fashion-MNIST and CIFAR10 when ensuring the convergence of model. The training hyperparameters of Deep SVDD (Ruff et al., 2018) and DROCC (Goyal et al., 2020) in table 6 and table 7 are both from the official settings and listed in Table 8. We do not obtain the time cost of OCGAN (Perera et al., 2019) with GPU acceleration since the official code occurs system error when using GPU.

Table 5: Training time(seconds) of one-class anomaly detection on MNIST without GPU acceleration.

| Normal Class | 0 | 1 | 2 | 3 | 4 | 5 | 6 | 7 | 8 | 9 | Avg |
|---|---|---|---|---|---|---|---|---|---|---|---|
| Deep SVDD (Ruff et al., 2018) | 1137.16 | 1254.45 | 1083.28 | 1160.93 | 1076.51 | 1021.4 | 1113.29 | 1152.54 | 1109.05 | 1097.22 | 1120.58 |
| OCGAN (Perera et al., 2019) | 18050.79 | 20400.09 | 17851.42 | 18579.87 | 17737.39 | 16850.2 | 17748.48 | 19853.55 | 17759.54 | 17751.16 | 18258.24 |
| DROCC (Goyal et al., 2020) | 21269.80 | 21134.64 | 22020.75 | 21157.20 | 22269.17 | 21159.76 | 21270.14 | 21014.52 | 20953.55 | 21346.44 | 21359.59 |
| RGP-GiHS (Ours) | 13665.78 | 15108.69 | 13227.82 | 12009.33 | 11697.65 | 11334.82 | 11883.15 | 12769.74 | 13086.94 | 11644.02 | 12642.80 |
| RGP-UiHS (Ours) | 14986.82 | 16898.84 | 12177.22 | 12136.23 | 12322.76 | 11029.81 | 12345.14 | 11726.78 | 11858.33 | 12115.39 | 12759.73 |
| RGP-UbHS (Ours) | 11766.71 | 13301.99 | 12132.35 | 12261.36 | 11678.36 | 10826.82 | 13657.7 | 15064.35 | 13863.64 | 14083.29 | 12863.65 |
| RGP-UoHS (Ours) | 14020.14 | 16666.39 | 14028.68 | 15276.65 | 13153.98 | 12812.56 | 14189.72 | 15797.74 | 13891.94 | 14330.59 | 14416.83 |

Table 6: Training time(seconds) of one-class anomaly detection on Fashion-MNIST with GPU acceleration.

| Normal Class | T-shirt | Trouser | Pullover | Dress | Coat | Sandal | Shirt | Sneaker | Bag | Ankle-boot | Avg |
|---|---|---|---|---|---|---|---|---|---|---|---|
| Deep SVDD (Ruff et al., 2018) | 563.46 | 561.48 | 566.92 | 561.12 | 565.27 | 557.55 | 563.02 | 559.02 | 564.22 | 560.34 | 562.24 |
| DROCC (Goyal et al., 2020) | 510.17 | 507.88 | 507.76 | 506.62 | 504.71 | 503.79 | 503.99 | 502.18 | 505.85 | 504.14 | 505.70 |
| RGP-GiHS (Ours) | 499.40 | 499.77 | 499.53 | 499.10 | 498.65 | 500.20 | 498.58 | 499.98 | 500.78 | 496.23 | 499.22 |
| RGP-UiHS (Ours) | 500.79 | 500.16 | 500.32 | 499.70 | 499.91 | 499.66 | 499.33 | 500.39 | 499.68 | 499.55 | 499.94 |
| RGP-UbHS (Ours) | 499.64 | 499.69 | 499.95 | 500.08 | 499.66 | 499.35 | 500.11 | 499.7 | 500.59 | 500.39 | 499.91 |
| RGP-UoHS (Ours) | 484.26 | 487.74 | 490.22 | 489.46 | 486.55 | 485.00 | 484.80 | 484.79 | 485.71 | 485.45 | 486.39 |

Table 7: Training time(seconds) of one-class anomaly detection on CIFAR10 with GPU acceleration.

| Normal Class | Airplane | Auto-mobile | Bird | Cat | Deer | Dog | Frog | Horse | Ship | Trunk | Avg |
|---|---|---|---|---|---|---|---|---|---|---|---|
| Deep SVDD (Ruff et al., 2018) | 943.97 | 946.16 | 941.57 | 932.88 | 944.01 | 943.51 | 942.04 | 942.93 | 941.69 | 941.69 | 942.04 |
| DROCC (Goyal et al., 2020) | 970.90 | 976.35 | 974.08 | 971.07 | 971.99 | 972.03 | 971.09 | 974.67 | 971.60 | 972.54 | 972.63 |
| RGP-GiHS (Ours) | 747.48 | 748.08 | 747.83 | 748.57 | 748.34 | 747.33 | 747.33 | 746.74 | 747.48 | 746.03 | 747.52 |
| RGP-UiHS (Ours) | 746.95 | 747.17 | 747.51 | 746.31 | 747.72 | 747.80 | 747.58 | 747.78 | 747.80 | 747.36 | 747.39 |
| RGP-UbHS (Ours) | 746.12 | 746.11 | 745.21 | 747.21 | 746.97 | 746.08 | 745.06 | 746.01 | 746.49 | 745.45 | 746.07 |
| RGP-UoHS (Ours) | 713.25 | 712.59 | 712.70 | 712.61 | 712.97 | 712.86 | 712.59 | 713.02 | 712.72 | 712.77 | 712.80 |

Table 8: Training settings of one-class anomaly detection on Fashion-MNIST, CIFAR10 and MNIST.

| Methods | Dataset | batch size | training |
|---|---|---|---|
| Deep SVDD (Ruff et al., 2018) | MNIST | 200 | 100(epochs) |
| DROCC (Goyal et al., 2020) | MNIST | 200 | 100(epochs) |
| OCGAN (Perera et al., 2019) | MNIST | 200 | 100(epochs) |
| RGP-GiHS (Ours) | MNIST | 200 | 100(epochs) |
| RGP-UiHS (Ours) | MNIST | 200 | 100(epochs) |
| RGP-UbHS (Ours) | MNIST | 200 | 100(epochs) |
| RGP-UoHS (Ours) | MNIST | 200 | 100(epochs) |
| Deep SVDD (Ruff et al., 2018) | Fashion-MNIST | 50 | 150(epochs) |
| DROCC (Goyal et al., 2020) | Fashion-MNIST | 256 | 100(epochs) |
| RGP-GiHS (Ours) | Fashion-MNIST | 200 | 100(epochs) |
| RGP-UiHS (Ours) | Fashion-MNIST | 200 | 100(epochs) |
| RGP-UbHS (Ours) | Fashion-MNIST | 200 | 100(epochs) |
| RGP-UoHS (Ours) | Fashion-MNIST | 200 | 100(epochs) |
| Deep SVDD (Ruff et al., 2018) | CIFAR10 | 50 | 150(epochs) |
| DROCC (Goyal et al., 2020) | CIFAR10 | 256 | 100(epochs) |
| RGP-GiHS (Ours) | CIFAR10 | 200 | 200(epochs) |
| RGP-UiHS (Ours) | CIFAR10 | 200 | 200(epochs) |
| RGP-UbHS (Ours) | CIFAR10 | 200 | 200(epochs) |
| RGP-UoHS (Ours) | CIFAR10 | 200 | 200(epochs) |

## B  PROOF FOR PROPOSITION 2.1

*Proof.* Letting $z_1, z_2, \ldots, z_d$ be i.i.d Gaussian variables with mean 0 and variance 1. According to Lemma 1 of (Laurent & Massart, 2000) (in which letting $a_1 = \cdots = a_D = 1$), the following

inequality holds for any $t$

$$\mathbb{P}\left(\sum_{i=1}^{d} z_i^2 \geq d + 2\sqrt{dt} + 2t\right) \leq \exp(-t). \tag{12}$$

Letting $\mathbf{z} = (z_1, z_2, \ldots, z_d)^\top$ and $r^2 = d + 2\sqrt{dt} + 2t$, we have

$$\mathbb{P}\left(\|\mathbf{z}\| \geq r\right) \leq \exp\left(-\frac{r^2 + \sqrt{d(2r^2 - d)}}{2}\right), \tag{13}$$

where $d < 2r^2$. This finished the proof. $\qquad\square$

## C   PROOF FOR PROPOSITION 2.2

*Proof.* Letting $z_1, z_2, \ldots, z_d$ be i.i.d variables of $\mathcal{U}(-1, 1)$. Then $\mathbb{E}(z_i) = \frac{1}{3}$ and $\mathrm{Var}(z_i) = \frac{4}{45}$, $i = 1, 2, \ldots, n$. Using Chebyshev's Inequality, for any $t > 0$, we obtain

$$\mathbb{P}\left(\left|\sum_{i=1}^{d} z_i^2 - \frac{d}{3}\right| \geq t\right) \leq \frac{4d}{45t^2}. \tag{14}$$

It follows that

$$\mathbb{P}\left(\sum_{i=1}^{d} z_i^2 \geq \frac{d}{3} + t\right) \leq \frac{4d}{45t^2}. \tag{15}$$

Letting $r^2 = \frac{d}{3} + t$, we have

$$\mathbb{P}\left(\|\mathbf{z}_i\| \geq r\right) \leq \frac{4d}{5(3r^2 - d)^2}, \quad j = 1, 2, \ldots, n. \tag{16}$$

Note that one may obtain tighter tail bound than (16). $\qquad\square$

## D   DETAILED NEURAL NETWORKS ARCHITECTURE

For the two image datasets (Fashion-MNIST and CIFAR-10), in our method, $f_\theta, g_\phi$ are both CNNs. The detailed neural network architecture is showed in Table 9.

| |
|---|
| $f_\theta$    in_channels=1, mid_dim=128, if Fashion-MNIST; in_channels=3, mid_dim=1024, if CIFAR-10. |
| Basic-Block(in_channels, out_channels=64) |
| Basic-Block(in_channels=64, out_channels=128) |
| Basic-Block(in_channels=128, out_channels=256) |
| Basic-Block(in_channels=256, out_channels=512), AveragePool2d |
| Linear(mid_dim, bias=False) |
| $g_\phi$ |
| Basic-Block(in_channels, out_channels=512) |
| Upsample(scale_factor=2, mode='nearest') |
| Basic-Block(in_channels=512, out_channels=256) |
| Upsample(scale_factor=2, mode='nearest') |
| Basic-Block(in_channels=256, out_channels=128) |
| Upsample(scale_factor=2, mode='nearest') |
| Basic-Block(in_channels=128, out_channels=64) |
| Basic-Block(in_channels=64, out_channels=32) |
| Conv2d(in_channels=32, out_channels=16, kernel_size=3, padding=1, bias=False) |
| BatchNorm2d(16), LeakyReLU |
| Conv2d(in_channels=32, out_channels=in_channels, kernel_size=3, padding=1, bias=False) |
| Basic-Block(in_channels, out_channels) |
| Conv2d(in_channels, out_channels, kernel_size=3, padding=1, bias=False) |
| BatchNorm2d(out_channels), LeakyReLU |
| Conv2d(out_channels, out_channels, kernel_size=3, padding=1, bias=False) |
| BatchNorm2d(out_channels) |

Table 9: Architecture of the CNN-based neural network for CIFAR-10 and Fashion-MNIST.

For the three tabular datasets (Abalone, Arrhythmia, Thyroid), in our method, $f_\theta, g_\phi$ are both MLPs. The detailed neural network architecture is showed in Table 10

| $f_\theta$ | $g_\phi$ |
|---|---|
| Linear(input_dim, 64, bias=False), LeakyReLU
Linear(64, 128, bias=False) | Linear(128, input_dim, bias=False) |

Table 10: Architecture of the MLP-based neural network for tabular dataset.

