# OpenReview forum: "Restricted Generative Projection for One-Class Classification and Anomaly detection"
_ICLR.cc/2023/Conference — Submitted to ICLR 2023_

### Official Review · Reviewer_23uq · 2022-10-24

**Confidence:** 5
**Clarity, Quality, Novelty And Reproducibility:** See "Strength and Weaknesses" section…
**Correctness:** 4
**Technical Novelty And Significance:** 2
**Empirical Novelty And Significance:** 2
**Recommendation:** 5

**Strength And Weaknesses:**

# Strength

* The paper is overall very clearly written. The mathematical notations are easy to follow. The key concepts, including the definition of outlier and the definition of Maximum Mean Discrepancy, are explained in detail.
* The problem this paper addresses is relevant to the machine learning community.
* The proposed algorithm shows promising empirical results.

# Weaknesses

* My biggest concern is that the proposed method looks significantly similar to Adversarial Autoencoder (AAE; Makhzani et al., 2015 ; https://arxiv.org/abs/1511.05644 ). AAE also uses MMD loss to encourage the mapped normal data to follow a specific distribution. Although the original AAE paper did not explore the outlier detection application, there are other papers that applied AAE to outlier detection tasks, such as [this](https://arxiv.org/abs/1902.06924).  Could you elaborate on the differences between the proposed method and AAE?

* How can we ensure that an outlier sample is always mapped to the outside of the designated region and does not fall into the inlier region in the latent space? Since the data space is high-dimensional and we often use a smaller dimensional latent space, we often observe "feature collapse" (for example, see [this paper](https://proceedings.neurips.cc/paper/2020/hash/543e83748234f7cbab21aa0ade66565f-Abstract.html) or [this paper](https://proceedings.mlr.press/v119/van-amersfoort20a.html)), meaning that an inlier A and an outlier B are mapped to the same latent space point Z.

* Having a good empirical performance on Table 1 and Table 2 are good, but the benchmark could have been made more challenging. An outlier detection task (and also a one-class classification problem as well, if you allow me to use the two terms interchangeably) tends to be more difficult as the variability within the inlier distribution becomes larger. For example, an outlier detection task where the inlier distribution consists of multiple clusters. This setting can be simulated by using nine out of ten classes as inliers and the remaining single class as outliers. [This paper (Yoon et al., 2021)](https://proceedings.mlr.press/v139/yoon21c.html) showed that autoencoder-based methods are particularly weak at this setting. The paper also introduced an autoencoder-based outlier detection method.

**Summary Of The Paper:**

This paper proposes an outlier detection algorithm that is based on bidirectional mapping between the data space and the latent space. The proposed algorithm maps the normal data to a restricted region in the latent space, letting the outliers map to the outside of the region.

**Summary Of The Review:**

The paper is generally clear and sound. The presented results seem reproducible. However, I do have a concern regarding its novelty, where I expect some comment from the authors.

---

### Official Review · Reviewer_sjwv · 2022-10-24

**Confidence:** 5
**Correctness:** 3
**Technical Novelty And Significance:** 2
**Empirical Novelty And Significance:** 1
**Recommendation:** 3

**Clarity, Quality, Novelty And Reproducibility:**

The paper is written well in general wit the exception of a few minor issues. The proposed method has some limited novelty, but the experimental evaluation is weak and the results are inferior to the state-of-the-art. The results can be reproduced if the authors share the source codes of the proposed methodology.

**Strength And Weaknesses:**

The main strengths of the paper can be summarized as follows:
i) In general, the paper is written well despite some minor errors.
ii) The authors propose some minor novelties over the existing methods in the literature.
iii) Better accuracies are obtained compared to some baseline methods.

The main weaknesses of the paper can be summarized as follows:
i) The novelty is very limited since the proposed method has close ties to the existing methods. More formally, it is very similar to auto-encoder networks in terms of architecture and loss function. Furthermore, compact hyperspheres and related distributions are used for target distribution, and these models are already used in Deep SVDD and its variants.
ii) Compact hypersphere models make sense for target distribution and it is largely used for anomaly detection. However, what is the motivation for using other target distributions such as Uniform between hyperspheres. What are the advantages?
iii) Experimental evaluation is quite weak and biased. The authors must compare their methods to the recent ones. Especially, Deep SVDD variants yield much higher accuracies. Please see the new references below. For example, the current SOTA AUC score is 96% on Cifar-10, and 93% on fashion Mnist datasets. The authors’ reported accuracies are much lower. Also, why do not the authors conduct tests on Mnist dataset (used in the most of the anomaly detection papers) and KDD and KDDNew datasets (used in Qiu, et al., Neural transformation learning for deep anomaly detection beyond images. In Proceedings of the International
Conference on Machine Learning, 2021)?
Minor Issues:
1) Please use inscribe or include instead of encase.
2) It seems there is typo in the first contribution. “… that are easy to be violated” is negative thing. It should be changed as “that are not easy to be evaluated”.
3) There is a mismatch between the equations (1) and (2). D_z should be changed as T(D_x)) in the first equation or the second equation must be corrected accordingly.

References:
[1] D. Hendrycks, M. Mazeika, T. Dietterich, Deep anomaly detection with outlier exposure, in: International Conference on Learning and Recognition (ICLR), 2019.
[2] P. Liznerski, L. Ruff, R. A. Vandermeulen, B. J. Franks, M. Kloft, K.-R. Muller, Explainable deep one-class classification, in: International Conference on Learning and Recognition (ICLR), 2021.
[3] L. Ruff, R. A. Vandermeulen, B. J. Franks, K.-R. Muller, M. Kloft, Rethinking assumptions in deep anomaly detection, in: International Conference on Machine Learning Workshops, 2021.
[4] L. Ruff, R. A. Vandermeulen, N. Gornitz, A. Binder, E. Muller, K.-R. Muller, M. Kloft, Deep semi-supervised anomaly detection, in: International Conference on Learning and Recognition (ICLR), 2020.
[5] I. Golan, R. El-Yaniv, Deep anomaly detection using geometric transformations, in: NeurIPS, 2018.


**Summary Of The Paper:**

In this paper, the authors propose a method for anomaly detection problems. To this end, the original data samples are transformed into a new space that has the desired target data distribution in the transformed space. The resulting loss function attempts to minimize both the learned distribution and the target distribution as well as the reconstruction errors of the original data samples. MLPs (multi-layer perceptron) are used to learn the transformation. The Gaussian in Hypersphere, Uniform hypersphere, Uniform between hyperspheres, and Uniform on a hypersphere models are used target distributions. The proposed method bears similarity to  auto-encoder networks (since exact architectures are used for learning) and Deep SVDD method since compact hypersphere are used for modeling target distributions. Therefore, the novelty is limited. The method is tested on several easy datasets and better accuracies are reported. However, the method is not compared to recent state-of-the-art methods. Therefore, comparisons are not fair.

**Summary Of The Review:**

The paper contribution is weak since there exists similar methods and there is no experimental evidence to demonstrate the superiority of the proposed method. More precisely, the results are inferior to the Deep SVDD variants that use compact hypersphere models for approximating normal data distributions as in the proposed method. Also, motivation for using other target distributions is not given. Therefore, my recommendation will be rejection.

---

### Official Review · Reviewer_ByPd · 2022-10-24

**Confidence:** 4
**Correctness:** 3
**Technical Novelty And Significance:** 2
**Empirical Novelty And Significance:** 2
**Recommendation:** 3

**Clarity, Quality, Novelty And Reproducibility:**

- The method is fairly simple and easy to follow.
- Injecting the introduced distribution priors is novel for AEs, but this can be regarded as a regularization term for AE, limiting the technical novelty of the paper.
- Most implementation details are included for reproducibility.

**Strength And Weaknesses:**

Strength
- The idea is simple and well-motivated.
- The paper is generally easy to follow.

Weaknesses
- In a nutshell, the method appears simply adding a regularization term for training an Auto-Encoder, though with a well-motivated distribution prior.
- Performance does not exceed the state-of-the-art in many cases (e.g., Tab 1 & 2).
- One model is needed for each class, making it less practical in a real deployment, for which a model that can handle all normal classes is preferred. Would it be possible to encode multiple classes in a single model for AD?
- How could the hyperparameter be determined without accessing the actual AD data?

**Summary Of The Paper:**

This paper addresses the abnormal detection problem by introducing a one-class AutoEncoder-based model, leveraging kernel maximum mean discrepancy (MMD) in the latent space. The model is expected to learn a bounded latent distribution for each class, requiring one AE model to be trained for each class. The model is trained with the MMD loss and an L2 reconstruction loss. The method is technically sound and good results are achieved on three tabular datasets and two image datasets.

**Summary Of The Review:**

Overall, the paper introduces a new AE-based method for abnormal detection by introducing distribution prior to the features in the latent space. The idea is interesting and generally effective, though not achieving state-of-the-art in many cases. The significance of technical contribution and results appears to be not strong enough for acceptance as a regular paper in ICLR, but perhaps a workshop.

---

### Official Review · Reviewer_wCV4 · 2022-10-24

**Confidence:** 3
**Correctness:** 4
**Technical Novelty And Significance:** 3
**Empirical Novelty And Significance:** 2
**Recommendation:** 6

**Clarity, Quality, Novelty And Reproducibility:**

The paper is clearly presenting the proposed method and the experiments are well conducted.

It seems that this work is novel; however, some connections are clear to other work in the literature such as VAE and the work of Perera et al. (2019)

**Strength And Weaknesses:**

See summary of review for more details

**Summary Of The Paper:**

This paper proposes a novel method for one-class classification and anomaly detection, by integrating restricted generative projection using 4 different variants of volume restrictions

**Summary Of The Review:**

The proposed methods is interesting. However, it seems that it is taking some ideas from different works and combining them, such as the used of MMD, as opposed to KL-divergence in VAE, and the use of hyperspheres with several variants, as opposed to a hypercube as proposed by Perera et al. (2019).

It is said that other methods may suffer from high computational cost and instability in optimization. However, this is also the case of the proposed method. The authors did not provide any analysis of the computational complexity of the proposed method. Moreover, the conducted experiments did not provide any measure of this computational complexity, such as the time requested for training, or computing the FLOPS.

There are many spelling and grammatical errors. We give here some of them:
The first category are, the second category are, the last category are
Hypershpere
all the samples distribute uniformly
discard all data points outsides
hypersphre
It was used to identity
Since both image datasets contains
we both set learning rate to
The details of the our network
under four different restriction

---

### Decision · Program_Chairs · 2023-01-20

**Decision:**

Reject

**Justification For Why Not Higher Score:**

N/A

**Justification For Why Not Lower Score:**

N/A

**Metareview: Summary, Strengths And Weaknesses:**

This paper addresses the problem of novelty detection by introducing a one-class AutoEncoder-based model, relying on kernel maximum mean discrepancy (MMD). The proposed algorithm maps the normal data to a restricted region in the latent space, letting the outliers map to the outside of the region.
Most reviewers agree that the paper is quite well written and easy to follow, that it addresses a relevant problem, and the motivation is interesting. However, the reviewers also expressed their concern regarding limited novelty since  the method is closely related to other auto encoder methods.
The paper suffers from some empirical deficiencies, The empirical results on small images are not that convincing as they do not generalize well to large images such as ImageNet. The description of the experimental protocol in applying the baseline is missing (e.g. choice of OC-SVM hyper parameters and other experimental decisions), which limit the credibility of these results.